# Sediment Transport near Ship Shoal for Coastal Restoration in the Louisiana Shelf: A Model Estimate of the Year 2017–2018

**Haoran Liu** [1,2,3,*,†]**, Kehui Xu** [1,2]**, Yanda Ou** [1,2]**, Robert Bales** [1,2]**, Zhengchen Zang** [4] **and Z. George Xue** [1,2,5]

1   Department of Oceanography & Coastal Sciences, Louisiana State University, Baton Rouge, LA 70803, USA; kxu@lsu.edu (K.X.); you3@lsu.edu (Y.O.); rbales1@lsu.edu (R.B.); zxue@lsu.edu (Z.G.X.)
2   Coastal Studies Institute, Louisiana State University, Baton Rouge, LA 70803, USA
3   Department of Experimental Statistics, Louisiana State University, Baton Rouge, LA 70803, USA
4   Department of Biology, Woods Hole Oceanographic Institution, Woods Hole, MA 02543, USA; zzang1@lsu.edu
5   Center for Computation and Technology, Louisiana State University, Baton Rouge, LA 70803, USA
*   Correspondence: cugharrison@gmail.com
†   Current address: 2172, ECE Building, Louisiana State University, Baton Rouge, LA 70803, USA.

**Abstract:** Ship Shoal has been a high-priority target sand resource for dredging activities to restore the eroding barrier islands in LA, USA. The Caminada and Raccoon Island pits were dredged on and near Ship Shoal, which resulted in a mixed texture environment with the redistribution of cohesive mud and noncohesive sand. However, there is very limited knowledge about the source and transport process of suspended muddy sediments near Ship Shoal. The objective of this study is to apply the Regional Ocean Modeling System (ROMS) model to quantify the sediment sources and relative contribution of fluvial sediments with the estuary and shelf sediments delivered to Ship Shoal. The model results showed that suspended mud from the Atchafalaya River can transport and bypass Ship Shoal. Only a minimal amount of suspended mud from the Atchafalaya River can be delivered to Ship Shoal in a one-year time scale. Additionally, suspended mud from the inner shelf could be transported cross Ship Shoal and generate a thin mud layer, which is also considered as the primary sediment source infilling the dredge pits near Ship Shoal. Two hurricanes and one tropical storm during the year 2017–2018 changed the direction of the sediment transport flux near Ship Shoal and contributed to the pit infilling (less than 10% for this specific period). Our model also captured that the bottom sediment concentration in the Raccoon Island pit was relatively higher than the one in Caminada in the same period. Suspended mud sediment from the river, inner shelf, and bay can bypass or transport and deposit in the Caminada pit and Raccoon Island pit, which showed that the Caminada pit and Raccoon Island pits would not be considered as a renewable borrow area for future sand dredging activities for coastal restoration.

**Keywords:** sediment transport; ROMS modeling; Ship Shoal; Caminada pit; Raccoon Island pit; coastal restoration

## 1. Introduction

Multi-resolution numerical models have been used in studies of coastal sediment modeling and morphological evolution in the Gulf of Mexico (GoM) in recent decades [1–3]. A dozen existing multidimensional models, both one-dimensional and three-dimensional, are here briefly described. For instance, the sediment module in Delft3D can be used to simulate the effect of sediment properties

on deltaic processes [4–6]. Caldwell and Edmonds [5] used DELFT3D to simulate the effect of sediment properties (the median, standard deviation, skewness, and percent of cohesive sediment) on the deltaic processes and morphology of the Mississippi River. Statistical modeling, including machine learning and deep learning, is also becoming popular in sediment modeling studies, especially in satellite image analysis and seafloor morphology identification [7–10]. Statistical methods provide a statistically optimal estimate by incorporating disparate data and manual interpretation, especially in locations that cannot be measured directly. Diesing et al. [7] tested the accuracy of different approaches, including geostatistics, image analysis, and machine-learning methods for acoustic data interpretation off the Scottish coast of the United Kingdom and found that the model performances were similar among the approaches. Liu et al. [9] tested multiple machine learning classifiers to identify the sediment types of the Caminada dredge pit in the eastern part of the submarine sandy Ship Shoal of the Louisiana inner shelf of the USA.

In recent decades, the regional ocean modeling system (ROMS) has been widely used in the oceanographic community to investigate the dynamics of ocean circulation [11–13]. This model has been incorporated in a coastal-circulation model with two-way coupling between a wave model and the sediment transport module [3,14]. Xu et al. [15] studied the seabed erosion and deposition on the Louisiana shelf in response to hurricanes Katrina and Rita in the year 2005 via 3D sediment models in ROMS and found that horizontal erosional patterns are mainly controlled by hurricane tracks and wave-current combined shear stresses. Moriarty et al. [16] applied a coupled hydrodynamic sediment transport-biogeochemistry mode in the northern GoM. Zang et al. adapted the coupled ocean-sediment transport model to the northern GoM over 20 years and found that decreased river discharge would largely affect the sediment concentration in waters around the delta [17].

In recent decades, the Louisiana coast has been facing extensive coastal land loss due to subsidence, the shortage of sediment supply from the Mississippi River, occasional hurricane landfalls, the frequent passages of winter storms, and human intervention such as dam construction and dredging navigation channels [18,19]. A major way to defend coastal land loss is to restore degraded barrier shorelines by dredging sand resources from borrow areas and delivering them to coastal sedimentary environments [20]. Although billions of cubic meters of sand are needed for initial and recurring restoration [21–25], high-quality sand is largely limited to isolated submarine shoals or infilled paleo-river channels on the inner shelf. Ship Shoal is estimated to contain 1.2 billion cubic meters of potential high-quality quartz sand [26]. Ship Shoal is considered the closest available sand resource for barrier shorelines in central Louisiana and has been dredged for significant volumes of the high-quality beach and dune restoration in Port Fourchon and the Grand Isle of Louisiana (Figure 1).

In recent years, several pits were dredged on and near Ship Shoal, including the Raccoon Island, Caminada, and Block 88 dredge pits. The Caminada restoration project was the first such project to use sand resources from the Ship Shoal area for barrier island restoration and represents the largest Louisiana monetary investment in restoration to date [25,27,28]. A total volume of $9.07 \times 10^6$ m$^3$ of sand was excavated from the Caminada pit based on pre- and post-construction surveys [27]. However, previous surveys suggest the existence of transient mud bypassing the seabed of Ship Shoal, which could fill borrow areas and affect physical/biological processes and sand quality for the future reuse of coastal restoration [9,29–33]. Stone et al. [25,31] hypothesized that occasional sediment plume shifts from the Atchafalaya Bay to the southeast might result in the accumulation of a thin fluid-mud layer on some portions of Ship Shoal. O'Connor et al. [34] found that the $^7$Be radioisotope activity in the muddy sediments collected inside Raccoon Island pit indicates a significant portion of the mud was derived from fluvial sources and deposited within ~6 months of core collection. Xue et al. [35] collected multiple corings in the Caminada pit and found that muddy sediment was deposited within six months from fluvial sources. Liu et al. [29] found that muddy sediments were transported into the Caminada pit of Ship Shoal and deposited in heterogeneous patches within two years of dredging. Liu et al. [9] applied multiple machine learning classifiers to identify sediment types and found mud was prone to deposit in the trough zones with lower backscatter values in the Caminada pit of Ship

Shoal. These findings together reveal that transient mud may temporarily blanket the sandy shoal but is later resuspended to fill in pits or is transported to deeper water. However, there is very limited knowledge about the source and transport process of suspended muddy sediments near Ship Shoal. Thus, the objective of this study is to apply the ROMS model in Ship Shoal to (1) quantify the sediment transport directions and fluxes near Ship Shoal, (2) compare the relative contribution of fluvial with the estuary and shelf sediments delivered to Ship Shoal, and (3) apply the model results in studies of dredge pits for future coastal restoration. Due to limitations in our current model resolution (250 m), dredge pits are not included in our model. Rather, the modeled "background" sediment concentrations near the pits are used to calculate the dredge pit infilling rates.

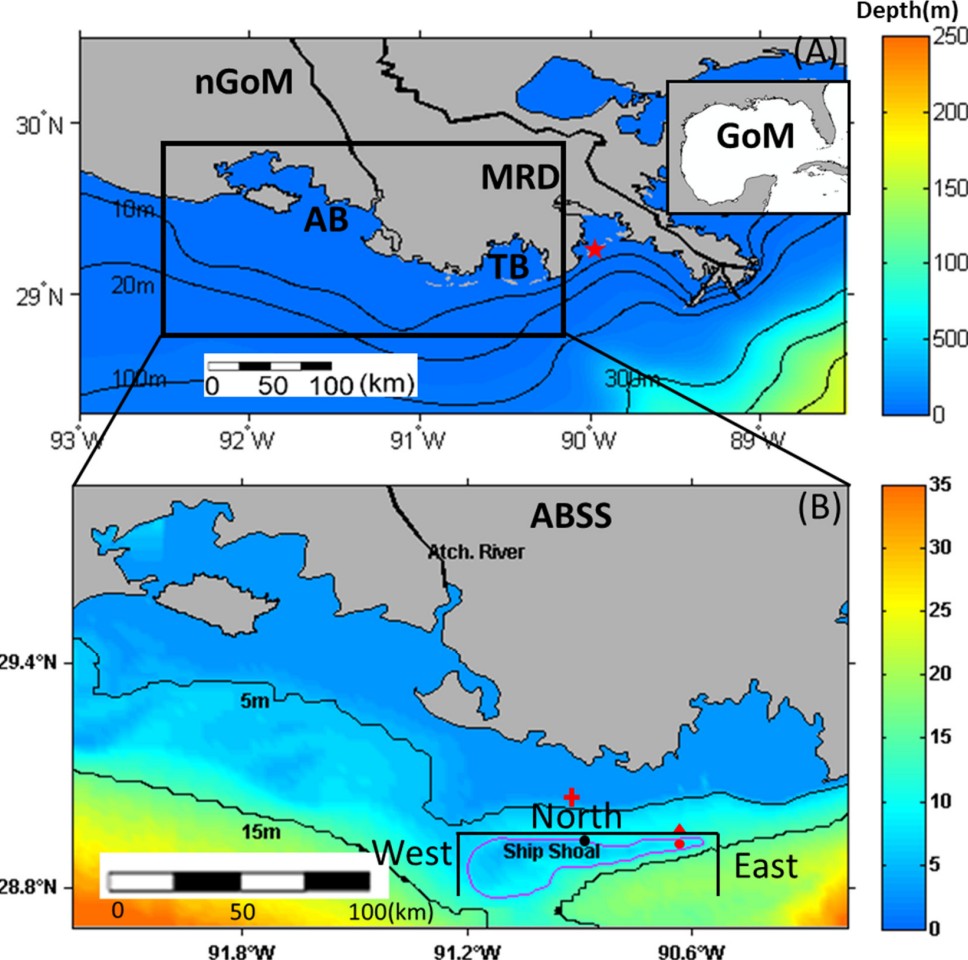

**Figure 1.** (**A**,**B**) The northern Gulf of Mexico (nGoM) and Atchafalaya Bay and Ship Shoal (ABSS) domains used in the Regional Ocean Modeling System (ROMS) overlaid with the water depth (color shading; ETOPO1), Grand Isle (red star), location of tripod observation (red triangle), Caminada pit (red circle), Raccoon Island pit (red cross), Block 88 pit (black circle), and three black transects (north, south, and east of Ship Shoal). (AB: Atchafalaya Bay; TB: Terrebonne Bay; MRD: Mississippi River Delta).

## 2. Model Setup

This study tests a series of numerical models using the Regional Ocean Modeling System, ROMS [13]. This three-dimensional, open-source model uses algorithms for an unlimited number of user-defined sediment classes and the evolution of the bed morphology [13]. ROMS can be incorporated in a coastal-circulation model with two-way coupling between a wave model and the sediment transport module [3,14]. The Community Sediment Transport Modeling System (CSTMS; https://woodshole.er.usgs.gov/project-pages/sediment-transport) is incorporated into the ocean model (ROMS) to simulate sediment transport and deposition. The bottom boundary layer, including

wave-orbital calculations and wave-current combined stress, was based on Styles et al. [36] and Harris et al. [37]. More details of the sediment transport module can be found in Warner et al. [13], Xu et al. [15], and Zang et al. [17].

The modeling period was 15 months, from 1 July 2017, to 1 October 2018, which was determined by the field surveying time of two geophysical field trips in the Caminada pit for the model validation (Figure 2). One tripod-attached wave gauge sensor was deployed in August 2017, and was used to calculate wave heights. Our model domain covered the Atchafalaya Bay and Ship Shoal (hereby defined as the ABSS domain) and had a 250 m horizontal resolution with 16 weighted vertical layers (Figure 1a). The northern Gulf of Mexico (nGoM) model developed by Zang et al. [17] was used to generate the east, south, and west boundary conditions for the ABSS domain. The initial conditions of current velocity, sea level, temperature, and salinity were also interpolated from Zang et al. [17]. The monthly average freshwater and suspended sediment inputs from rivers debouching into the GoM were retrieved from the United States Geological Survey (USGS)'s Water Data for the Nation website (http://nwis.waterdata.usgs.gov) and applied as the boundary conditions. The mesh bathymetry was interpolated and smoothed from the ETOPO1 dataset (https://www.ngdc.noaa.gov/mgg/global/). The wave data were generated by the Simulating Waves Nearshore model (SWAN, version 41.01) and then fed into ROMS as the wave forcing file. The model outputs were saved every 12 h for further analysis. The time step for the ABSS domain was 20 s.

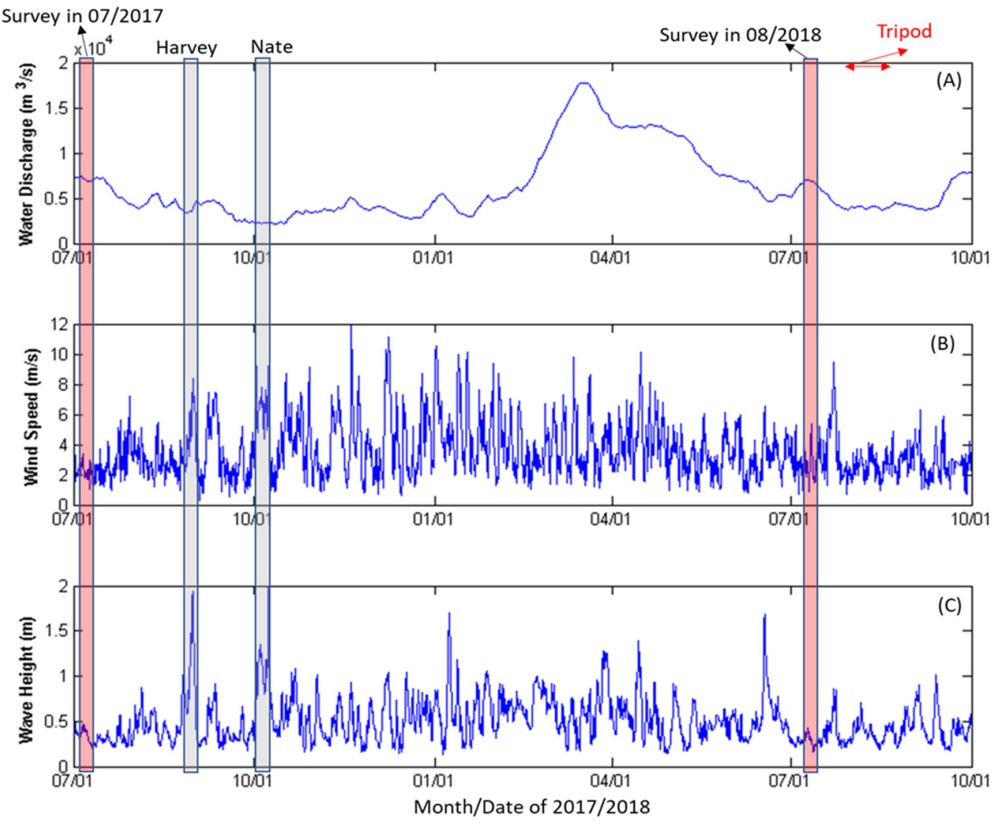

**Figure 2.** Time-series results in the Ship Shoal from 1 July 2017 to 1 October 2018. (**A**) Daily river discharge at United States Geological Survey (USGS) station 07374525 Atchafalaya River in Simmesport, LA. (**B**) Daily wind speed from National Oceanic and Atmospheric Administration(NOAA) station GISL1-8761724 in Grand Isle, LA. (**C**) Wave height from the Simulating Waves Nearshore (SWAN) model calculated at the tripod (see the location in Figure 1). Two geophysical surveys were completed in July 2017 and August 2018, marked as red blocks, respectively. Hurricanes Harvey (17 August 2017–1 September 2017) and Nate (4 October 2017–8 October 2017) and a cold front (1 December 2017) are highlighted in grey blocks.

Zang et al. [17] used four cohesive and two non-cohesive sediment classes for the river inputs and two cohesive and non-cohesive sediment classes for the seabed. As this study focuses on the sediment exchange between areas shallower and deeper than 5 m isobath, two shallow-sediment and two deep-sediment classes are added for the seafloor based on the 5 m water depth contour (Figure 3). The sand content of the seabed is interpolated from the USGS's usSEABED database [38], following the method of Xu et al. [39]. We set the initial sediment concentration in the water column as zero on 1 January 2017, and then we ran the model for six months until 1 July 2017 to reach a relatively stable condition.

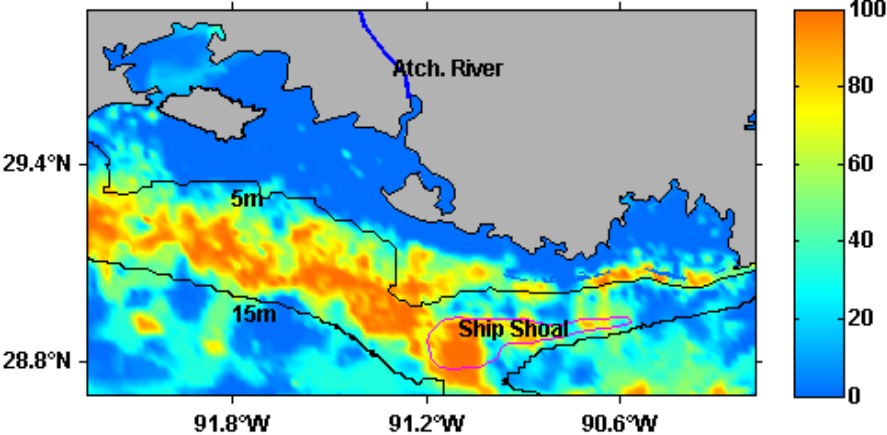

**Figure 3.** Interpolated sand proportion from the usSEABED data, added contours of 5 and 15 m. Shallow-sand and deep-sand classes are classified by the 5 m bathymetry contour. Polygon of Ship Shoal is marked in red and considered as the reference.

To achieve the most reasonable sediment parameterization, we used many parameters from Xu et al. [15,39] and Zang et al. [17] and compared our simulation results for surface suspended sediment concentration (SSC) against the map derived from the Moderate Resolution Imaging Spectroradiometer. In Table 1, we list a summary of the sediment model parameterization used in this study. We prescribed four layers of sediment on the seafloor, each with a thickness of 1.0 m. This study generated a total of 10 classes of sediment tracers. Four classes of mud and two classes of sand from the Mississippi River and Atchafalaya River are added into this model. The seabed is classified into four classes as shallow sediments and deep sediments to explore the sediment exchange between shallow and deep water. The seabed erosion–deposition was based on non-cohesive parameterizations [13]. The SSC at the boundaries of the ABSS domain is interpolated from nGoM, and this study applies gradient and radiation boundary conditions to avert unreal artificial sediment plumes along the boundaries.

**Table 1.** Sediment characteristics parameterization.

| Sediment Type | Grain Diameter (mm) | Settling Velocity (mm/s) | Critical Shear Stress (Pa) | Erosional Rate ($10^{-4}$ kg/m$^2$/s) |
|---|---|---|---|---|
| Mud_01(Mississippi River) | 0.004 | 0.1 | 0.1 | 5 |
| Mud_02(Mississippi River) | 0.03 | 0.1 | 0.16 | 5 |
| Mud_03(Atchafalaya River) | 0.004 | 0.1 | 0.1 | 5 |
| Mud_04(Atchafalaya River) | 0.03 | 0.1 | 0.16 | 5 |
| Mud_05(Shallow mud) | 0.004 | 0.1 | 0.1 | 5 |
| Mud_06(Deep mud) | 0.004 | 0.1 | 0.1 | 5 |
| Sand_01(Mississippi River) | 0.0625 | 1 | 0.2 | 5 |
| Sand_02(Atchafalaya River) | 0.0625 | 1 | 0.2 | 5 |
| Sand_03(Shallow sand) | 0.0625 | 1 | 0.2 | 5 |
| Sand_04(Deep sand) | 0.0625 | 1 | 0.2 | 5 |

## 3. Model Validation

### 3.1. Hydrodynamics Model Validation

The ABSS domain is interpolated and nested in the nGoM domain. For a wave, this study gathers the daily averaged significant wave height at the tripod station in Ship Shoal (Figure 1b). The model-data comparison reveals a very good agreement between the simulated and observed significant wave height ($R^2$ = 0.84; Figure 4). To further evaluate the model simulated salinity over a longer period, this study interpolated the simulated salinity to the observation sites at corresponding periods and compared it against the available measurements from the Southeast Area Monitoring and Assessment Program (http://seamap.gsmfc.org), which has a total of 48 data points from the surfaces, middle, and bottom of the water column in 16 stations, covering the period from 2017 to 2018 (Figure 5). The model-observation comparison in Figure 5b indicates that the ocean model is capable of reproducing the pattern of salinity distribution ($R^2$ = 0.69), with low-salinity water embracing Atchafalaya Bay and coastal Louisiana over the inner shelf and high-salinity water further offshore.

### 3.2. Sediment Model Validation

For the sediment model, we compare the simulated surface SSC against the map derived from the Moderate Resolution Imaging Spectroradiometer (MODIS-aqua; Figure 6). We select two nearly cloud-free satellite images in September 2017 and April 2018 and apply them to the ABSS domain using the SSC algorithm developed by Miller and McKee [40]. In September 2017, turbid water from the Atchafalaya River dominated the entire Atchafalaya Bay and coastal water (water depth < 15 m). Westward sediment transport (black arrows in Figure 6a) could be detected over the coastal Chenier Plain, where the westward alongshore current was strong (Figure 6a). In April 2018, both the SSC and spatial scale of the sediment plume increased dramatically due to the high river discharge. The water discharge from the Atchafalaya River reached the highest level in March 2018 (Figure 2a), which expanded sediment plume toward the south (Figure 6c). The westward transport along the Chenier Plain coast reduced when compared with September 2017. During high river discharge, the sediment plume from the Atchafalaya River could reach Ship Shoal (Figure 6d). Although the magnitudes of SSC are different between the model and satellite data in April 2018, they generally agree on the spatial pattern. For instance, in both September 2017 and April 2018, the SSC is generally high inside and south of both Atchafalaya Bay and Terrebonne Bay, and moderately high in the inner shelf (0–5 m deep) due to resuspension by the high near bed wave orbital velocity. The SSC, however, is low in water deeper than 15m in both the model and satellite data. The difference between the model result and satellite image in the shallowest parts of inner bays was likely due to: (1) the low waves generated inside the bay near land, and (2) the fact that marsh-edge erosion and exchange are not considered in the model. The missing satellite data for the inner bay (grey zones inside bay) were due to the presence of dense clouds, sun glint, and water vapor in the coastal region.

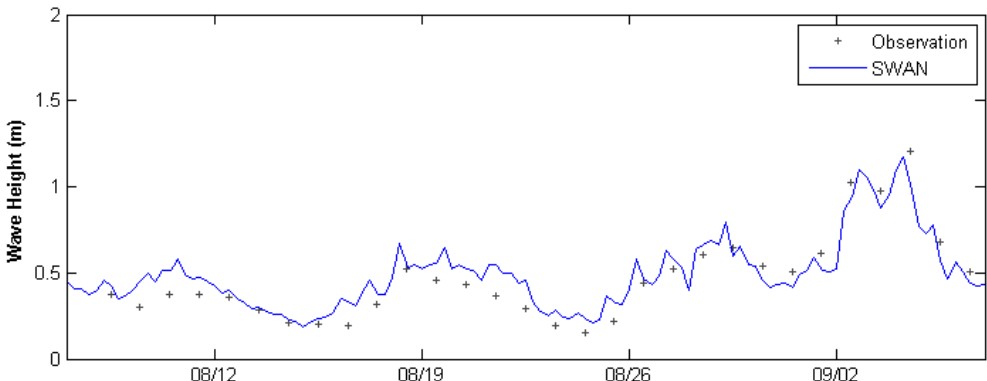

**Figure 4.** Comparison between the observed and modeled daily mean significant wave height at the tripod station from 8 July 2018 to 17 September 2018. See the location of the tripod in Figure 1.

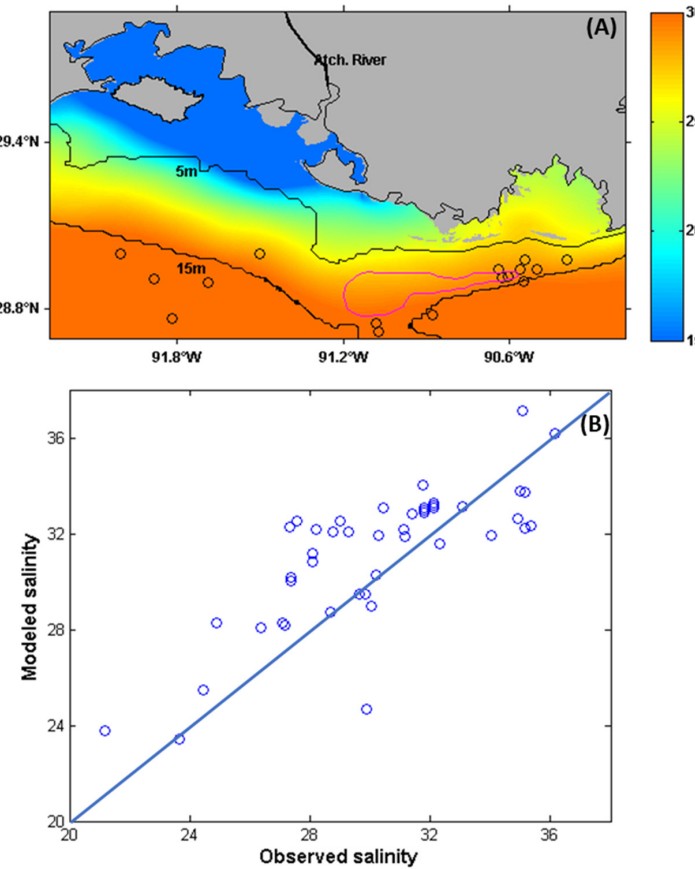

**Figure 5.** (**A**) Salinity comparison between the observations (data source: SEAMAP; http://seamap.gsmfc.org/ and tripod sensors); (**B**) Model results for the time-averaged surface salinity in the ABSS domain from July 2017 to October 2018. See the stations of salinity observation in A. For each station, we interpolate the model results to the locations of the stations to ensure comparability ($R^2 = 0.69$).

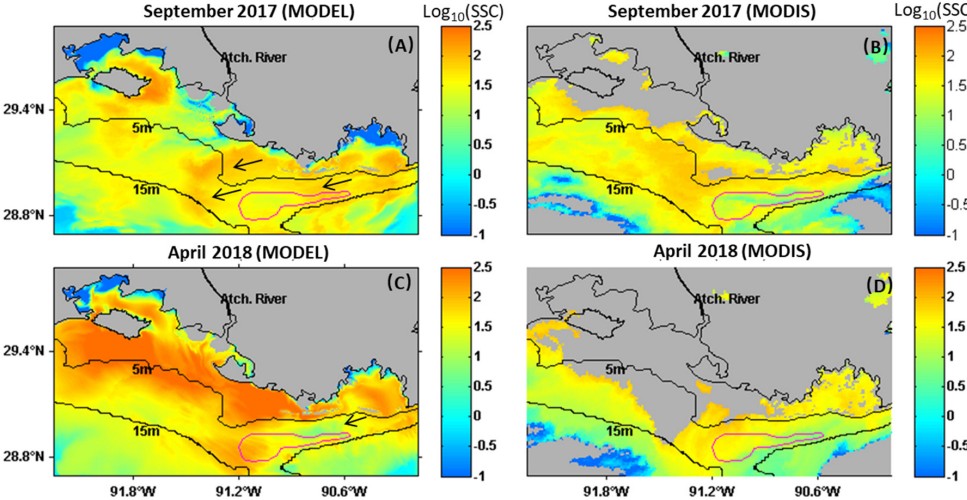

**Figure 6.** (**A**) 8-day averaged model-simulated surface suspended sediment concentration (SSC) in September 2017 (11 September 2017–19 September 2017). (**B**) The Moderate Resolution Imaging Spectroradiometer (MODIS) (aqua)-derived surface SSC in September 2017 (11 September 2017–19 September 2017). (**C**) 8-day averaged model-simulated surface SSC in April 2018 (16 April 2018–24 April 2018). (**D**) The MODIS (aqua)-derived surface SSC April 2018 (16 April 2018–24 April 2018), during which the number of good quality satellite images (no sun glint and cloud-free) is the largest. Unit is in log10 (mg/L). Black arrows indicate the sediment transport directions.

## 4. Results

### 4.1. Averages for the Year 2017 to 2018

For the period from 1 July 2017 to 1 October 2018, the time-averaged surface salinity from the ROMS model was low near the mouth of Atchafalaya River, which increased southwestward from the river mouth (Figure 7a). The depth-averaged currents moved southwestward at 0.1–0.2 m/s, which was relatively strong on the shelf and then changed to be weaker offshore. The wave orbital velocity was high near Ship Shoal and south of the barrier island near Terrebonne Bay (Figure 7b). The peak wave height in the model domain varied from 0.5 (from 5 m isobath) to 1 m (15 m isobath) (Figure 7c). Ship Shoal is located in a region with a relatively shallow depth and tall waves, which led to a higher orbital velocity. In the mouth of Terrebonne Bay, the wave magnitude was high compared with the inner and northern part of the bay, which showed as the elongated narrow yellow bar (see the dash line in Figure 7b). Two small tidal inlets inside Terrebonne Bay also showed a high orbital velocity (Figure 7b). The time-averaged and depth-integrated suspended fluvial sediment in the water column was estimated to be high close to the mouths of the Atchafalaya River (Figure 7d). As shown in Figure 7d, the year-average sediment concentration in ABSS varied greatly from 100 mg/L to almost zero. Most suspended sediment from the Atchafalaya River was confined in the inner shelf to bay mouth. The sediments deposited landward of the 5 m isobath but a small portion of the surface sediments could transport to Ship Shoal within the 15 m isobaths (Figure 7d).

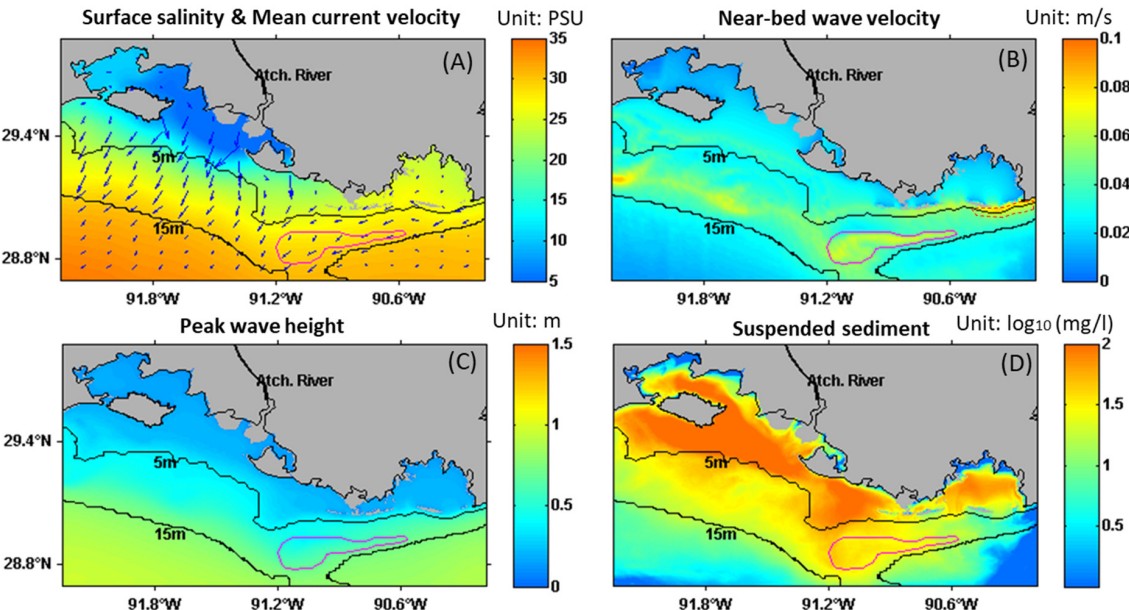

**Figure 7.** (**A**) Time-averaged surface salinity (PSU) and mean current velocity (m/s) calculated for the year 2017 and 2018. (**B**) Near-bed wave orbital velocity (m/s). (**C**) Peak significant wave height (m) estimated by SWAN. (**D**) Time-averaged and depth-integrated fluvial suspended sediment (mg/L in logarithmic scale) in the water column calculated for the year 2017 and 2018.

### 4.2. Sediment Dispersal

The seabed sediment mass can be determined from the thickness, grain size distribution, porosity, grain density, and others [41]. This study averaged the model-simulated mud mass from 1 July 2017 to 1 October 2018 based on the changes in the sediment mass. The changes in the seabed mud mass during this period show the sediment transport from Atchafalaya River to the bay and then to the inner shelf of coastal Louisiana. As shown in Figure 8a, the mud accumulated on the seabed was high near the mouth of the Atchafalaya River and then decreased in the offshore direction to the inner shelf. Large amounts of the Atchafalaya sediments were retained inside the bay. Over the western

Louisiana shelf, fluvial sediments were transported westward, crossing 91.2° W, and were deposited over the shelf (Figure 8a). From 1 July 2017 to 1 October 2018, a limited amount of sediment from the Atchafalaya River moved southeastward, passed Ship Shoal, and accumulated southeast of Ship Shoal (green, about 1 kg/m$^2$). Still, the negligible amount can be preserved on top of Ship Shoal (blue, about 0.01 kg/m$^2$). Previous studies hypothesized that occasional sediment plume shifts from the Atchafalaya Bay to the southeast might result in the accumulation of a thin fluid–mud layer on some portions of the shoal [9,31,35]. Our model results verify and prove that sediment passing above Ship Shoal is a possibility. Suspended mud may temporarily blanket the shoal or inside Caminada pit but is later transported elsewhere. The sediment transport process between the bay and shelf exchange was dynamic in our model domain (Figure 8b). The suspended mud from water shallower than 5m could transport to even cross Ship Shoal, which is also considered as the primary sediment source infilling the Caminada dredge pit (Figure 8b). Over the simulation period, our model shows that minimal fluvial sediment was deposited in waters deeper than 10 m, indicating limited cross-shelf suspended sediment transport.

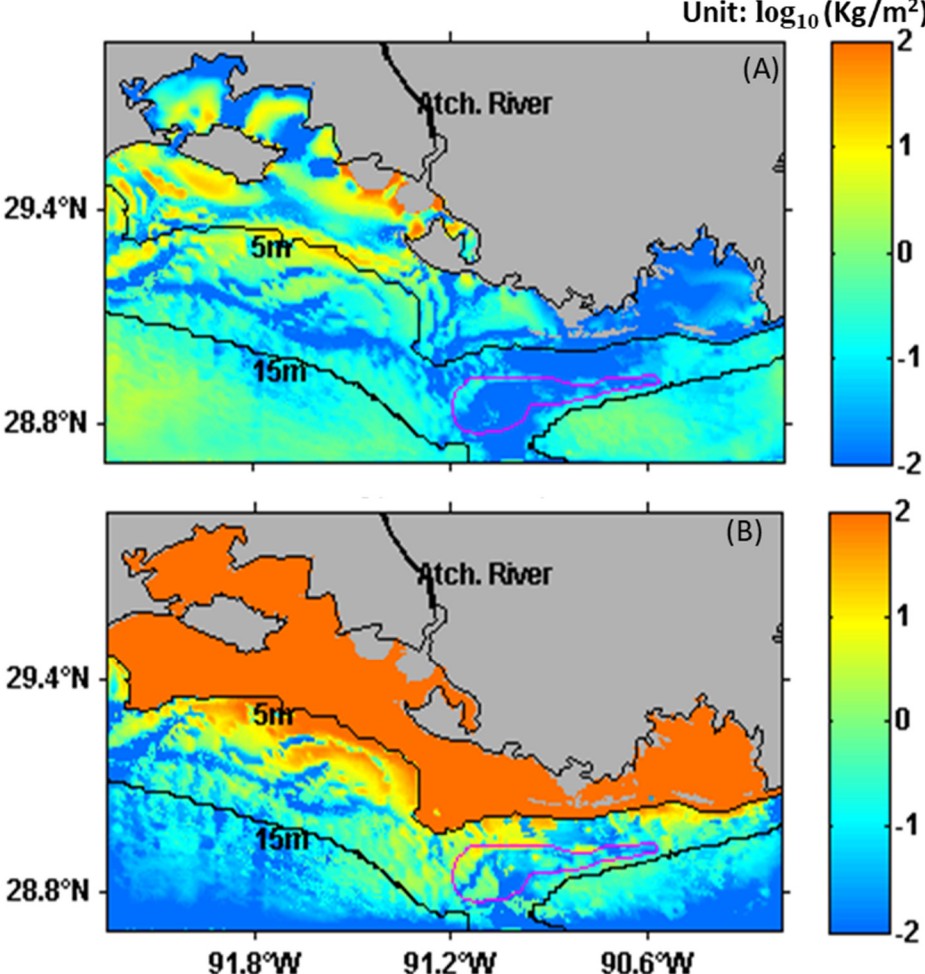

**Figure 8.** Change in the seabed mud mass in the ABSS domain from 1 July 2017 to 1 October 2018. (**A**) Sum of mud_03 and mud_04 in the log10 scale from the Atchafalaya River, and (**B**) mud_05 in the log10 scale from the bay and inner shelf shallower than 5 m. Mud_03 are mud_04 are two sediment tracers from the Atchafalaya River. Mud_05 is the shallow mud tracer showing in Table 1. The unit is log$_{10}$ (Kg/m$^2$). The magenta line delineates the boundary of the Ship Shoal dredging pit.

## 5. Discussion

### 5.1. Sediment Transport Flux near Ship Shoal

The sediment transport fluxes were calculated along three transects (see the locations in Figure 1b) north, east, and west of Ship Shoal, respectively. The magnitudes of the east and west sediment fluxes of all 10 sediment tracers generally exceeded those of the northward fluxes (Figure 9b,c). The negative magnitude in the north transect indicated a southward sediment flux from shallow areas in the bay and inner shelf to Ship Shoal (Figure 9a). Though the sediment flux in the west and east transects was mainly to westward (Figure 9c), sometimes the winds reversed, switching to eastward flux (Figure 9b). During Hurricane Harvey (17 August 2017–1 September 2017) in 2017, there was a high eastward flux of 5 kg/m/s with high wind speed (8 m/s) and high wave height (2 m) (Figure 2b,c). The eastward fluxes restarted once the winds changed, leading to eastward sediment transport for the storm. After that, Hurricane Nate (4 October 2017–8 October 2017) and a cold front (1 December 2017) led to an increased westward flux. It is well known that the hurricanes are counter-clockwise in the northern hemisphere. Since Ship Shoal was ~300 km east of the track line of Hurricane Harvey (Figure 10), its sediment transport was eastward. However, Ship Shoal was ~200 km west of the track line of Hurricane Nate, and its sediment transport was westward (Figure 10). This indicates hurricanes could bring sediments to infill dredge pits in Ship Shoal. However, the contributions from hurricanes are highly dependent on the distance between the hurricane track lines and dredge pits and the orientations and categories of hurricanes.

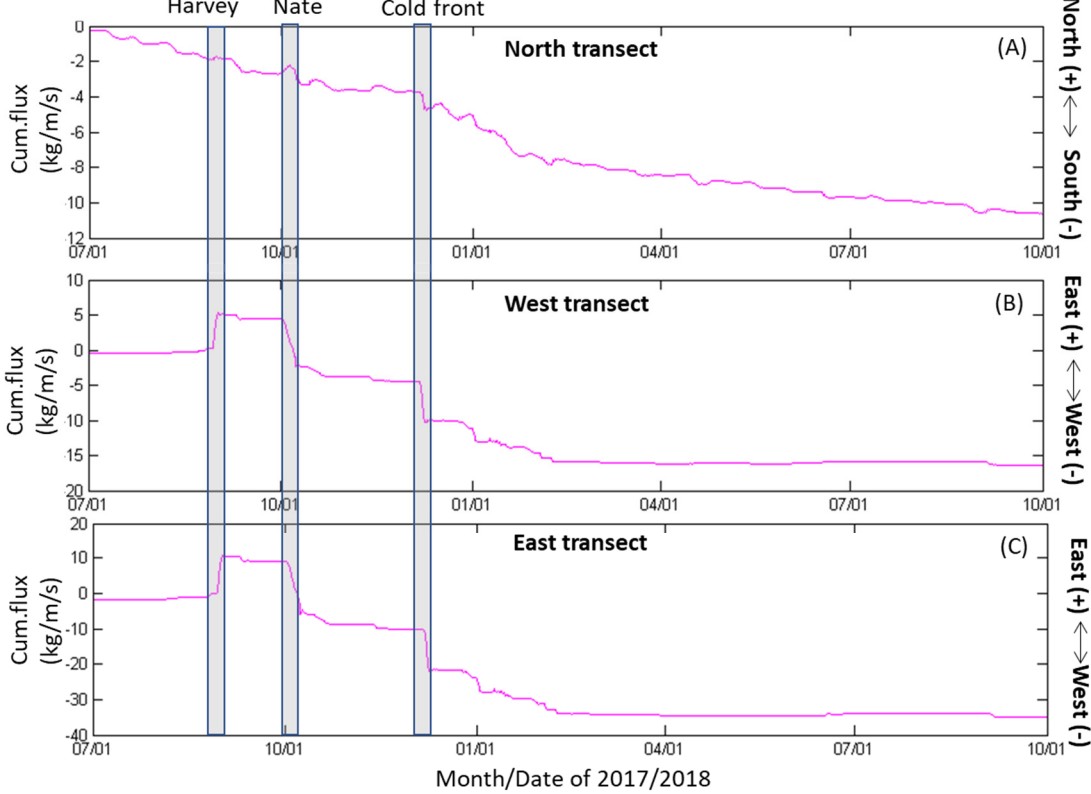

**Figure 9.** Cumulative sediment fluxes (in kg/m/s) near the north (**A**), west (**B**), and east (**C**) transects. See the locations of transects in Figure 1b.

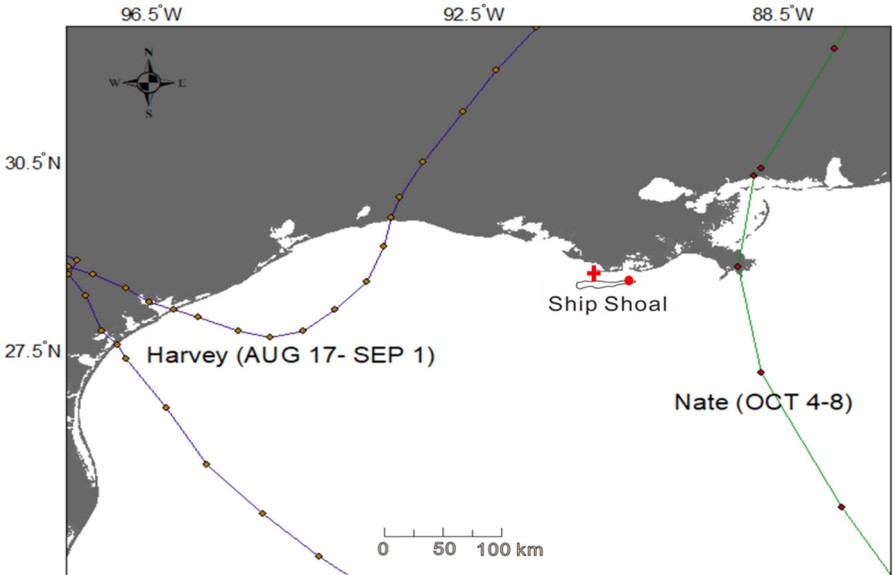

**Figure 10.** Hurricane and tropical storms track from 2017 to 2018 near Ship shoal. Data on the tracks of hurricanes Harvey and Nate were downloaded from the NOAA national hurricane center (https://www.nhc.noaa.gov/). The locations of the Caminada pit (red circle) and Raccoon Island pit (red cross) are shown in the figure.

*5.2. Application to Dredging Studies for Coastal Restoration*

In recent years, several pits were dredged in the ABSS domain, including the Raccoon Island, Caminada, and Block 88 dredge pits (Figure 1). The calculation of sediment thickness infilling dredge pits is critical for future dredging activities. Ribberink and Nairn [42] applied the 1D analytical approach and modeled the evolution of a proposed dredge pit in Block 88 (see the location in Figure 1b), which is 31 km west of the Caminada pit. Their equation has the following form:

$$\Delta Z_b = K_1 C_0 \omega_s T \frac{1}{\rho_{dry}} [1 - (\frac{h_0}{h_1})^3],$$

(1)

where $\Delta Z_b$ is the total siltation thickness per tide (m) and $K_1$ is the empirical coefficient. $C_0$ is the background concentration outside the pit, which is generally determined using the tide-mean and depth-averaged sediment concentration for the surrounding area (kg/m$^3$ or mg/L). $\omega_s$ is the settling velocity of mud (m/s). T is the tidal period (s). $\rho_{dry}$ is the dry bulk density (kg/m$^3$). $h_0$ is the water depth outside the pit (m). $h_1$ is the water depth inside the pit (m).

Their equation included the empirical coefficients, the settling velocity of mud, the water depth inside and outside the pit, the background concentration outside, and the tidal period, but lacked the consideration of wave resuspension, episodic extreme hurricanes, and other tropical events. It should be noted that they used a constant value for sediment concentration. Due to proximity, it is assumed that the Caminada and Raccoon Island pits have a depositional environment similar to the Block 88 pit. Due to the model grid horizontal resolution of 250 m, our ABSS model domain did not include the Caminada and Raccoon Island pits in which sediments can be trapped (Figure 8). Instead of using one constant in Nairn's model [42], we derived a time series of the bottom sediment concentrations ($C_0$) inside the Caminada and Raccoon Island pits based on the ABSS model output from 1 July 2017 to 1 October 2018, and kept the other parameters the same.

From our calculations, the sediment concentration inside the Caminada pit was very low (near 0 mg/L) in most of the days during the modeling period (Figure 11a). However, the sediment thickness per day infilling the Caminada pit increased to 0.005 m and even 0.017 m during the hurricanes or cold fronts that occurred during our modeling period (Figure 11b). In contrast, the sediment

concentration inside the Racoon Island pit was much higher than in the Caminada (0.05 mg/L) pit, and the sediment infilling in the Raccoon Island pit was modeled to be dynamic and episodically increased with storm events (Figure 11c). These findings for the sediment infilling rate from modeling matched previous results from sediment coring and geophysical observation in the Caminada pit and Raccoon Island pit. Xue et al., for instance, analyzed the $^7$Be penetration depths in a repeat multi-coring study in the Caminada pit and found seasonal sedimentation rates ranging between 7.3 and 55.0 cm/year in 2017 [35]. Liu et al. found the sediment infilling rate in the Caminada pit of 15.00 cm/year calculated by a repeat bathymetric survey in 2017–2018 [43]. Similarly, O'Connor [34] found that the $^7$Be profiles from the Raccoon Island pit accumulation rates exceeded an average of 0.24 cm/day, and that the multicores within the pit contained almost entirely silt. Our model also captured similar patterns, in which the bottom sediment concentration in the Raccoon Island pit was generally higher than it was in Caminada during the same period. The total calculated sediment thickness in the Raccoon Island pit from July 2017 to October 2018 (~2.1 m) was more significant than in the Caminada pit (~0.25 m) in the ABSS model domain (Figure 11).

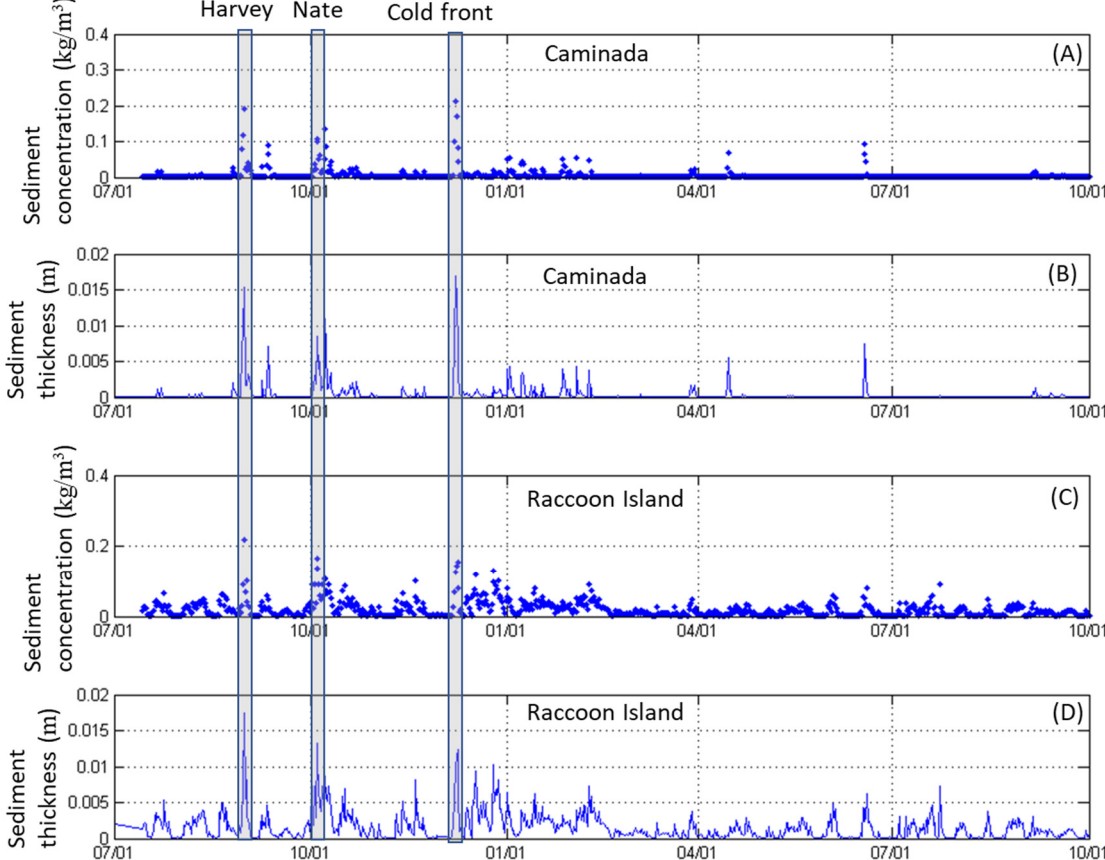

**Figure 11.** Time series change in sediment thickness in the Caminada pit (**A**,**B**) and Raccoon Island pit (**C**,**D**) from 7 July 2017 to 1 October 2018. Two tropical events and one cold front are marked in grey blocks.

Our modeling results show that hurricanes Harvey and Nate and one cold front in 2017 suspended sediment, which can contribute to the infilling of dredge pits (Figure 11), but they accounted for less than 10% of the sediments infilling both the Caminada and Raccoon Island dredge pits from July 2017 to October 2018. Xu et al. [15] reported that the region to the east of the tracks of hurricanes Katrina and Rita had stronger winds, taller waves, and deeper erosions at the centimeter to meter level. The relatively low contribution of 10% should be mainly due to very large distances (>100s km) from the dredge pits to hurricanes Harvey and Nate. If the eyes of the hurricanes had passed the pits within a few kilometers, their contribution would be presumably much higher (up to 40 cm) [17].

Besides, hurricanes can generate very tall waves that trigger landslides and the mass wasting of pit walls, which are not in our model. Moreover, hurricane contributions of 10% should be used with caution because of the following simplifications and assumptions: (1) the interaction between dredge pits and the hydrodynamic conditions (including waves and currents) is neglected; (2) the wind input for our ROMS model may not be high enough to fully resolve hurricanes (Harvey and Nate) when dealing with high spatial resolutions; and (3) right now our ROMS model only included the suspended sediment load but did not include any bed load or fluid mud processes, which can be significant contributors to pit infilling.

Xue et al. [35] found that the laminations from sediment coring were related to the Hurricane Harvey-induced deposition in 2017. The depth-average bottom concentration from our model correlated well with the occurrence of Hurricane Harvey, Hurricane Nate, and the cold front in 2017 (Figure 11). It indicated that hurricanes or tropical storms were potential sources infilling the Caminada and Raccoon Island pits. However, Nairn's model, shown above, was not included in this factor. Additionally, Nairn's model applied a constant sediment concentration when calculating the infilling process. Still, our modeling result indicated that the sediment concentration was temporally dynamic inside the Caminada pit and Raccoon Island pit (Figure 11). The future modeling work of the infilling process should consider these factors.

Regarding the sediment budgets for coastal restoration in Louisiana, the longshore transport in our model domain derived sediment coming from the river, inner shelf, and bay westward to Texas (Figure 8). It will lead to decreasing/losing sediment, especially during strong long-shore currents. Suspended mud sediment from the river, inner shelf, and bay can bypass or transport and deposit in the Caminada pit, which indicates that the Caminada and Raccoon Island pits would not be considered as renewable pits for future sand dredging activities for coastal restoration.

*5.3. Comparison with Other Worldwide Major Rivers*

Wright et al. [44] recognized at least four stages in the dispersal of sediment seaward from six global large river mouths, including supply via plumes, initial deposition, resuspension and transport, and long-term net accumulation. Walsh et al. [45] analyzed the dispersal-system type of more than 100 river mouths and found four dispersal systems, which include proximal-accumulation-dominated (PAD), marine-dispersal-dominated (MDD), estuarine-accumulation-dominated (EAD), and subaqueous-delta-clinoform (SDC) systems (Table 2). Delaware river, for instance, is classified as an EAD system because most of the river sediments accumulate within the estuary with the flocculation and estuarine circulation process, but the load of the river system is sufficiently small that the estuary remains unfilled [45]. MDD systems are characterized by bottom-boundary layer transport and sediment gravity flows, and most of the fine-grained sediment load cannot accumulate near the river mouth due to the energetic receiving-basin conditions, such as Columbia river [46]. The Ganges, Amazon, and Yangtze river mouths are classified as SDC systems due to strong tide flow effects, in which tidal currents could carry the fine-sediment load far away from the river mouths [47–49]. Similar to the Mississippi river, which is a PAD dispersal system [45], Atchafalaya River can be regarded as a distributary of the Mississippi system and shows a system with a low oceanographic energy and minimal resuspension after the initial depositional process [50].

Our model results show that most of the sediment from Atchafalaya River remains near the loci of the initial deposition near the river mouth and Atchafalaya bay (Figure 8a). This typical PAD system in Atchafalaya River, however, can behave like some other MDD rivers. Previous studies found that much of the suspended sediment from the Mississippi and Atchafalaya rivers is shunted offshore of the modern Balize delta mouth and the younger Wax Lake and Atchafalaya deltas [51], save for isolated outlets in Breton Sound and Barataria basin [52]. Finer sediment via resuspension can be transported and spread more uniformly along isobaths, expanding the Atchafalaya buoyant plumes by even transport across Ship Shoal (Figure 8b), and this process is similar to MDD systems.

Our study highlights that many large and small rivers may behave in contrasting modes and stages in the sediment dispersal process.

**Table 2.** Characteristics of the dispersal systems and classifications of multiple larger rivers.

| River | Drainage Basin Area (km$^2 \times 10^6$) | Water Discharge (km$^3$/year) | Sediment Load (Mt/year) | Wave Height/Tide Range (m) | Shelf Width (km) | Sediment Dispersal Stage | References |
|---|---|---|---|---|---|---|---|
| Delaware | 0.03 | 11.7 | 1 | 1.2/1.7 | 133 | EAD | [44] |
| Columbia | 0.67 | 251 | 15 | 2.9/2.5 | 49 | MDD | [46] |
| Ganges | 1.08 | 378 | 1060 | 0.9/3.6 | 321 | SDC | [47] |
| Amazon | 6.15 | 6300 | 1200 | 1.1/4.8 | 330 | SDC | [48] |
| Yangtze | 1.94 | 900 | 480 | 1.1/3.4 | 557 | SDC | [49] |
| Mississippi | 327 | 580 | 400 | 0.9/0.3 | 15 | PAD | [18] |
| Atchafalaya | 0.006 | 315 | 0.54 | 0.5/0.2 | 140 | PAD + MDD | [39,45] |

*5.4. Limitations and Future Work*

This simulation reproduced the overall pattern of sediment dispersal and bay shelf exchange in the ABSS. However, it is noteworthy that a few critical sediment transport mechanisms were not included in our model. First, sand and mud transport in a different way, and for future studies the comparison of these two will be tested. Secondly, this study does not consider wave-supported fluid mud in the ABSS domain. Previous studies [53,54] found that wave-supported fluid mud movement is another important mechanism in terms of fluvial sediment across-shelf transport over the muddy Atchafalaya Shelf. Lastly, the marsh edge sediment inside bays is not considered in this model. It is likely to lead to a low sediment concentration near the marsh edge. For future modeling work, the ABSS domain can be subdivided into a small domain with high-resolution grid nested dredge pits inside. It could be better to test and compare the sediment infilling process inside pits and outside pits. This study only uses the sediment module in ROMS with wave feeding as the input, and future studies could be focused on the coupling of SWAN and ROMS.

**6. Conclusions**

This study adapted a sediment transport model to the ABSS to investigate the sediment dynamics and exchange from the riverine/bay/inner shelf to Ship Shoal. Our model simulation shows that:

(1) Suspended fluvial sediment in the Atchafalaya River accumulated close to the river mouth and was confined to the continental shelf and bay mouth. Only a small portion of the sediments can transport to Ship Shoal within 15 m isobaths. The coastal current carried some Atchafalaya sediment westward.

(2) Suspended mud from the Atchafalaya River can transport and pass above Ship Shoal. The sediment transport process between the bay and shelf exchange was dynamic in the ABSS. Suspended mud from the inner shelf could also be transported to even cross Ship Shoal and generate a thin mud layer, which was also considered as the primary sediment source infilling the Caminada dredge pit.

(3) Since Ship Shoal was ~300 km east of the track line of Hurricane Harvey, its sediment transport was eastward. Ship Shoal was ~200 km west of the track line of Hurricane Nate, and its sediment transport was westward. Two hurricanes and one cold front changed the direction of the sediment flux near Ship Shoal, even though the distance between the hurricanes and Ship Shoal was more than 200 km. This indicated that hurricanes could bring a proportion of the sediment infilling pits in Ship Shoal. However, the hurricanes contributed less than 10% of the sediment infilling in both the Caminada and Raccoon Island dredge pits from July 2017 to October 2018 due to the

great distances involved. This percentage can change dramatically for other hurricanes during other periods.

(4) Our model also captured that the bottom sediment concentration in the Raccoon Island pit was relatively higher than the one in Caminada in the same period. The total sediment thickness in the Raccoon Island pit (~2.3 m) was greater than that in the Caminada pit (~0.25 m) in the ABSS model domain. Suspended mud sediment from the river, inner shelf, and bay can bypass or transport and deposit in the Caminada pit and Racoon Island pit, which showed that the Caminada and Raccoon Island pits would not be considered as renewable pits for future sand dredging activities for coastal restoration.

**Author Contributions:** The individual contributions to this manuscript are listed below: conceptualization, H.L. and K.X.; methodology, K.X., Z.Z. and Z.G.X.; software, Y.O., Z.Z. and R.B.; validation, H.L. and R.B. formal analysis, H.L. and K.X.; investigation, K.X. and R.B.; resources, Y.O. and Z.Z.; data curation, K.X. and R.B.; writing—original draft preparation, H.L.; writing—review and editing, K.X. and Z.G.X.; visualization, Y.O.; supervision, K.X. and Z.G.X.; project administration, K.X.; funding acquisition, K.X. All authors have read and agreed to the published version of the manuscript.

**Funding:** Funding for this study was provided by the U.S. Department of the Interior, Bureau of Ocean Energy Management, Coastal Marine Institute, Washington DC, under Cooperative Agreement Numbers M16AC00018 and M17AC00019.

**Acknowledgments:** We are thankful to Bingqing Liu from the Water Institute of the Gulf for sediment concentration validation via satellite images. The first author's Ph.D. program was founded by Economic Development Assistantship from the Graduate School of Louisiana State University. We are grateful to the Editors and multiple anonymous reviewers who provided valuable comments and suggestions.

**Conflicts of Interest:** The authors declare no conflict of interest.

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
