# Peer review of "Sediment Transport near Ship Shoal for Coastal Restoration in the Louisiana Shelf: A Model Estimate of the Year 2017–2018"

_water, doi:10.3390/w12082212_

Round 1

Reviewer 1 Report

See attach file.

Figures 1, 7, 9 and 10 must be improved

Author Response

Re: Reviewer 2

  1. Figures 1

We added the location of block 88 in figure 1 as the black dot.

  1. Figure 7

We added unit for surface salinity and mean current velocity.

  1. Figure 9

We add N-S or W-E location reference at the ends of the transects.

  1. Figure 10

We added locations of Caminada and Raccoon Island pits on this map.

Reviewer 2 Report

The paper presents a numerical investigation to quantify sediment transport near Ship Shoal. The manuscript is consistent with the aim of the special issue, however, the paper presents solely a case study and therefore it is too mainly addressed to local managers and stakeholders since neither a more general methodology nor a comparison with other case studies are presented. The following comments and remarks should be considered to improve the quality of the work.

Major:

1) The introduction should be revised:

1a) The objectives of the study should be immediately presented. It should be clarified that the paper concerns solely the analysis of a case study and therefore it does not describe a methodology applicable elsewhere. Unfortunately, in my opinion, this kind of paper does not appeal to an international reader.

1b) There are too many references in the first lines (17 in the first 18 lines of the introduction) and the reason for their citation is not clarified. For instance: "Statistical modelling, including machine learning and deep learning, is also becoming popular in the sediment modelling studies, especially in the satellite image analysis and seafloor morphology identification [7-10]". In my opinion, this method of citing everything together does not give value to references. I suggest distinguishing the citation [9] concerning the case study analyzed in the present manuscript. Similar considerations apply to citations [1-3], [4-6].

2) In the “Model setup” paragraph, I suggest adding a more detailed description of the ROMS model and a reference.

3) In Figure 6a and 6c, arrows indicating the current direction should be added in order to easily follow the sediment model validation description. Moreover, it is not clear why “Modeling and satellite data generally agreed well on spatial patterns“, especially for the comparison in April 2018. The authors correctly explain some reasons for differences; however, this sentence should be revised adding additional finding of the agreement.

Minor:

4) Please be consistent along with all the manuscript on how to mention the references. For instance, in line 50 “…combined shear stresses [15].” vs in line 50 “Moriarty et al. (2018) applied a coupled…”.

5) The caption of Figure 2 has a mistake. If I understand correctly, the colour of the blocks mentioned for surveys and hurricane are reverse.

6) In line 183, the correct cited figure is probably 6C instead of 6D.

Author Response

Re: Reviewer 1

The paper presents a numerical investigation to quantify sediment transport near Ship Shoal. The manuscript is consistent with the aim of the special issue, however, the paper presents solely a case study and therefore it is too mainly addressed to local managers and stakeholders since neither a more general methodology nor a comparison with other case studies are presented. The following comments and remarks should be considered to improve the quality of the work.

Major:

1) The introduction should be revised:

1a) The objectives of the study should be immediately presented. It should be clarified that the paper concerns solely the analysis of a case study and therefore it does not describe a methodology applicable elsewhere. Unfortunately, in my opinion, this kind of paper does not appeal to an international reader.

We added  a new Section 5.3 for the comparison with other worldwide major rivers. Previous studies reported the different sediment dispersal stages for each of the major rivers, such as Ganges, Amazon, Yangtze, and Mississippi rivers. This manuscript analyzed the sediment dispersal stage of Atchafalaya river and identified it as mixed stages of proximal-accumulation-dominated (PAD)and marine-dispersal dominated (MDD). Our model shows most of the sediment from Atchafalaya river deposits near the loci of initial deposition near the river mouth and Atchafalaya bay, which is characterized as a PAD dispersal system. It shows a system with low oceanographic energy and minimal resuspension after the initial depositional process. However, finer sediment via resuspension can be transported and spread more uniformly along isobaths, expanding Atchafalaya buoyant plumes by even transport across Ship Shoal, which process is like MDD systems.

This section illustrates that knowledge gained from this study can be applied to other river system worldwide. It also help scientists better understand the complexity of river dispersal systems. Many large and small mountainous rivers behave in multiple ways and stages.

1b) There are too many references in the first lines (17 in the first 18 lines of the introduction) and the reason for their citation is not clarified. For instance: "Statistical modelling, including machine learning and deep learning, is also becoming popular in the sediment modelling studies, especially in the satellite image analysis and seafloor morphology identification [7-10]". In my opinion, this method of citing everything together does not give value to references. I suggest distinguishing the citation [9] concerning the case study analyzed in the present manuscript. Similar considerations apply to citations [1-3], [4-6].

Added as reviewer suggested. We added details of some citations showing the process of sediment transport modeling in the Gulf of Mexico.

Caldwell and Edmonds [5] used DELFT3D to simulate the effect of sediment properties (the median, standard deviation, skewness, and percent cohesive sediment) on deltaic processes and morphology of the Mississippi River. Statistical modeling, including machine learning and deep learning, is also becoming popular in the sediment modeling studies, especially in the satellite image analysis and seafloor morphology identification [7-10]. Statistical methods provide a statistically optimal estimate by incorporating disparate data and manual interpretation, especially in the locations that cannot be measured directly. Diesing et al. [7] tested the accuracy of different approaches including geostatistics, image analysis, and machine-learning methods for acoustic data interpretation in Scottish coast of the United Kingdom and found the model performances were similar. Liu et al. [9] tested multiple machine learning classifiers to identify the sediment types of Caminada dredge pit in the eastern part of the submarine sandy Ship Shoal of Louisiana inner shelf of USA.

2) In the “Model setup” paragraph, I suggest adding a more detailed description of the ROMS model and a reference.

Added as the reviewer suggested.

This project tests a series of numerical models using Regional Ocean Modeling System, ROMS (Haidvogel et al., 2008; http://www.myroms.org/). This three-dimensional, open-source model implements algorithms for an unlimited number of user-defined sediment classes and the evolution of the bed morphology [13]. ROMS incorporated in a coastal-circulation model with two-way coupling between a wave model and the sediment transport module [3, 14]. The Community Sediment Transport Modeling System (CSTMS; http://www.cstms.org/) is incorporated into the ocean model (ROMS) to simulate sediment transport and deposition. The bottom boundary layer including wave-orbital calculations and wave-current combined stress were based on Styles et al. [36] and Harris et al. [37]. More details of the sediment transport module can be found in Warner et al. [13] and Zang et al. [17].

3) In Figure 6a and 6c, arrows indicating the current direction should be added in order to easily follow the sediment model validation description. Moreover, it is not clear why “Modeling and satellite data generally agreed well on spatial patterns“, especially for the comparison in April 2018. The authors correctly explain some reasons for differences; however, this sentence should be revised adding additional finding of the agreement.

We added arrows in Figure 6 to show the sediment current directions. We also added sentences in text to explain this. Although the magnitudes of SSC are different between model and satellite data in April 2018, they generally agree on the spatial pattern. For instance, in both Sep 2017 and Apr 2018, SSC is generally high inside and south of Atchafalaya Bay and Terrebonne Bay, and moderately high in the inner shelf (shallow water 0-5m deep) due to resuspension by high near bed wave orbital velocity. SSC, however, is low in water deeper than 15m on both model and satellite data.

We admit that we did not do extensive sensitivity tests on parameters like settling velocity, critical shear stress and erosion rate. 1) Our model grid is 900x500 and the computational time of 1-year model run is very expensive and is beyond the scope of our study. 2) Xu et al. (2011 and 2015) and Zang et al. (2019) have done extensive work on the parameterization and model validation. 3) The lead author graduated from LSU and will start working and our project funded by Bureau of Ocean Energy Management ended a few months ago. We believe that future tuning of parameters will match model with the satellite data better but that is our future work.

Minor:

4) Please be consistent along with all the manuscript on how to mention the references. For instance, in line 50 “…combined shear stresses [15].” vs in line 50 “Moriarty et al. (2018) applied a coupled…”.

Changed as reviewer suggested.

5) The caption of Figure 2 has a mistake. If I understand correctly, the colour of the blocks mentioned for surveys and hurricane are reverse.

Changed as reviewer suggested.

6) In line 183, the correct cited figure is probably 6C instead of 6D.

Changed as reviewer suggested.

Round 2

Reviewer 2 Report

Thank you for your edits and replies in response to my comments. I now find the manuscript suitable for publication.

Author Response

Re:

Dear Authors
according to the latest reply by the reviewers, I'm glad to inform you that your paper is accepted. Please consider to provide some minor check/modifications or correct few typos:

1) the format of the references in incomplete. Need to be expanded with all the author list in the proper format (see Instruction for Authors)

I downloaded reference style Guide for MDPI-Endnote and updated all my references.

2) check the alignment of figures and related captions in the same page

Changed as reviewer suggested.

3) Equation n. 1 need to be inserted immediately after its citation in the text (line 294) in a manner like "Their equation has the following form: _____________ (1), then the text can continue as in lines (304-308). Soon after it continues with lines 296-303 beginning with "The approach lacked consideration...."

Changed as reviewer suggested for this equation.

4) Typos/corrections in lines:
- line 83: 7Be needs to be reported in correct isotopic (7 upper script) form
- line 236: sediment-class --> consider to change in "grain-size"
- line 243: check the degree position in the text to respect the form 91.2° W.
- line 307: h0 and h1 to report with 0 and 1 lower script
- line 361: supple? to check (maybe supply)
- line 384: Table 1 is correctly Table 2. Please consider to modify mt/y since not clear the term "m". If millions, better the use of 10^6 t (6 upper script) or M (capital).

Changed all these typos